Mapping species of greatest conservation need and solar energy potential in the arid Southwest for future sustainable development

Fleckenstein Kylee kmflecke@asu.edu
Stein Adam
Bateman Heather L.
De Albuquerque Fabio
School of Applied Sciences and Arts, College of Integrative Sciences and Arts, Arizona State University , Mesa , AZ , United States of America
Sunny Armando
Electronic publication date: 2025 Jan 2
Publication date: 2025
Volume: 13
Electronic Location ID: e18568
Received 2024 Aug 15; Accepted 2024 Oct 31
Copyright: ©2025 Fleckenstein et al.
Copyright year: 2025
Copyright holder: Fleckenstein et al.
License: This is an open access article distributed under the terms of the Creative Commons Attribution License, which permits unrestricted use, distribution, reproduction and adaptation in any medium and for any purpose provided that it is properly attributed. For attribution, the original author(s), title, publication source (PeerJ) and either DOI or URL of the article must be cited.
License URL: https://creativecommons.org/licenses/by/4.0/

Keywords: Renewable energy, Species distribution, Species conservation, Climate change, Spatial analysis

Funding: ASU Foundation This project was supported by funding provided to ASU through a donation from the Ørsted Project to the ASU Foundation. The funders had no role in study design, data collection and analysis, decision to publish, or preparation of the manuscript.

==============================
The need for renewable energy has become increasingly evident in response to the climate change crisis, presenting a paradoxical challenge to biodiversity conservation. The Southwest United States is desirable for large-scale solar energy development (SED) due to its high global horizontal irradiance (GHI) values and vast open landscapes. However, this region is also rich in unique ecological and biological diversity. Several distinct species have garnered special attention as human population growth, habitat alteration, and climate change have accelerated in recent decades (i.e., LeConte’s Thrasher (Toxostoma lecontei), Bendire’s Thrasher (Toxostoma bendirei), Sonoran Desert Tortoise (Gopherus morafkai), Mojave Desert Tortoise (Gopherus agassizii), and the Southwestern population of the Burrowing Owl (Athene cunicularia). As the United States prepares to increase its development in renewable energies, particularly solar energy, there has been a growing concern about how this development will further impact these species. In this study, we propose a novel combined approach to find areas of high habitat suitability for endangered species within areas of high SED potential. Specifically, we employed species distribution modeling (SDM) to identify areas with suitable habitats and likely species presence, and we conducted a site suitability analysis for potential SED locations within the Southwest. As a result, we found significant overlap between potential SED locations and the high-priority habitats of all target species, thus underlining the importance of prioritizing conservation efforts as more solar projects are reviewed in these Southwestern states. Our study aims to inform conservationists and developers in making sustainable decisions for the region’s future development.

Introduction

As countries adapt to the escalating climate crisis, the urgency to transition toward renewable energy sources has become more apparent. Among the various renewable energy options, solar energy is a highly promising and practical solution (Devabhaktuni et al., 2013). However, if done improperly, solar energy development (SED) can negatively impact important conservation areas and threaten biodiversity (Thomas et al., 2018; Rehbein et al., 2020). Currently, about 13% of peer-reviewed journal articles currently mention or focus on the importance of renewable energy siting, highlighting the growing attention to balancing energy development with ecological considerations (Agha et al., 2020). Therefore, identifying where SED potential and conservation priorities overlap is essential for minimizing ecological harm.

Identifying suitable locations for large-scale SED require project-specific considerations; however, several key variables consistently emerge across solar suitability studies. Global horizontal irradiance (GHI)—the amount of solar radiation received by a horizontal surface—is essential for maximizing photovoltaic power generation (Gbémou et al., 2021; Nzelibe, Ojediran & Moses, 2022). GHI is of particular interest for assessing the viability of solar projects (Gao et al., 2024). Another critical factor is minimal vegetative land cover, as areas with dense vegetation, wetlands, or forests can hinder solar development and create conflicts with conservation efforts (Doljak & Stanojević, 2017; Mierzwiak & Calka, 2017; Bukhary, Ahmad & Batista, 2018; Nebey, Taye & Workineh, 2020). Recent studies further emphasize the importance of avoiding densely vegetated areas, prioritizing open landscapes to reduce ecological impacts (Loquias et al., 2022).

In addition, limited slope is necessary for the efficient alignment and installation of solar panels. Slopes beyond certain thresholds can complicate installation and increase project costs (Charabi & Gastli, 2011; US Department of Energy, 2015; Alami Merrouni, Mezrhab & Mezrhab, 2016; Sabo et al., 2017; Rodrigues, Coelho & Pedro, 2017). Proximity to infrastructure, such as substations further enhance the feasibility of solar projects by minimizing energy losses and reducing the costs of new infrastructure (Katkar et al , 2021; Farthing et al., 2016).

Lastly, prioritizing land outside protected areas is crucial to prevent conflicts with biodiversity conservation and ensure alignment with environmental goals (Thomas et al., 2018; Nzelibe, Ojediran & Moses, 2022). Recent studies have emphasized the importance of aligning renewable energy development with conservation goals. For example, Cameron, Cohen & Morrison (2012) demonstrated that solar projects sited on lower conservation-value lands, such as degraded agricultural areas can potentially reduce ecological harm while advancing the renewable energy objectives. Similarly, Walston et al. (2018) further expanded on this idea by showing that prioritizing degraded lands not only supports solar development but can create opportunities for pollinator and native plant recovery. Though these studies provide essential broad-scale insights into the relationship between solar development and environmental conservation, they do not address the specific intersection of habitat suitability and renewable energy potential—a key focus of this study.

The United States Southwest (hereafter Southwest, Fig. 1) has been identified as a key region for solar development as it is a part of the Western Solar Plan (Bureau of Land Management Solar Program Environmental Impact Statement, 2023), which ranks among the highest in the United States for GHI (Agha et al., 2020; Sengupta et al., 2018). In addition, the region’s arid and semi-arid climates and the availability of flat and unobstructed terrains make it desirable for installing utility-scale solar power plants (Prăvălie, Patriche & Bandoc, 2019).

Figure 1 Study extent of the United States Southwest.

Study extent includes California, Nevada, Colorado, Arizona, New Mexico, and Texas.

However, the Southwest has also been recognized as a “hotspot” for threatened and endangered species within the United States (Flather, Knowles & Kendall, 1998). It is home to several arid-adapted species identified as “Species of Greatest Conservation Need”, whose general habitat requirements resemble the characteristics associated with high solar development potential (Arizona Game and Fish Department, 2022). Among the endangered species, the following are of critical interest: the LeConte’s Thrasher (Toxostoma lecontei), Bendire’s Thrasher (Toxostoma bendirei), Sonoran Desert Tortoise (Gopherus morafkai), Mojave Desert Tortoise (Gopherus agassizii), and the Southwestern population of the Burrowing Owl (Athene cunicularia).

The impacts of SED on ecosystems and biodiversity are primarily related to habitat loss and alteration, both recognized as major threats to biodiversity (Tsoutsos, Frantzeskaki & Gekas, 2005; Gasparatos et al., 2017; Hernandez et al., 2014; Thomas et al., 2018). Landscape changes caused by SED extend beyond the solar facilities themselves, encompassing supporting infrastructure such as access roads and equipment, which can result in an altered area approximately 2.5 times larger than the footprint of the panels (Gasparatos et al., 2017). Another threat is bird and bats mortality caused by collisions with solar panels or transmission lines, especially when they mistake panels for water sources (Bennun et al., 2021; Karban et al., 2024).

In the Southwest, many public areas are being evaluated for SED. These include areas with high biodiversity value that include protected species (Lovich & Ennen, 2011). Areas with high potential for SED often conflict with arid ecosystems with exceptional biodiversity, especially the Chihuahuan, Sonoran and Mojave deserts (Parker, Cohen & Moore, 2018; Lovich & Ennen, 2011). These deserts have high GHI potential and are already facing adverse impacts of SED, including changes in plant communities and habitat use (Karban et al., 2024). Another threat of SED to the Southwest deserts is the change in hydrology and water availability, a high amount of water is often needed to clean the solar panels (Bennun et al., 2021). The withdrawal of water in arid lands is linked to adverse impacts on riparian ecosystems (e.g., plant density reduction, Webb & Leake, 2006). Furthermore, the construction and decommissioning of SED facilities may lead to the destruction and modification of wildlife habitat, with soil disturbances acting as pathways for invasive species, potentially out-competing native ones (Turney & Fthenakis, 2011; Gasparatos et al., 2017; Lovich & Ennen, 2011; Moore-O’Leary et al., 2017). Finally, large-scale solar energy infrastructure can act as both a physical and visual obstruction, disrupting the natural movement patterns of some species. Additionally, solar installations can create heat island effects, altering local temperatures and further stressing wildlife (Karban et al., 2024). These risks highlight the need for targeted mitigation strategies to avoid ecological harm and ensure sustainable development.

Ignoring the potential ecological impacts of SED could result in negative consequences for threatened species and ecosystems, leading to potential legal challenges, public opposition, and interventions that could jeopardize future operations (Turney & Fthenakis, 2011). With the rapid shift towards renewable energy, particularly in the Southwest, developers must thoroughly assess and consider the potential ecological and biological impacts on biodiversity (Thomas et al., 2018; Sekercioglu et al., 2011). Mitigation strategies, although available, can be mostly species-specific, costly, and less effective (Agha et al., 2020; Phalan et al., 2017). Therefore, conservation planning tools such as those highlighted by Kreitler et al. (2015), emphasize the importance of predictive analyses to guide development while minimizing biodiversity conflicts. By identifying sites where high SED potential and priority habitat overlap, developers can identify priority areas for conservation that can be avoided, minimize ecological harm, and ensure sustainable development through this proactive approach.

Herein, we propose a combined approach to find areas of high habitat suitability for endangered species within areas of high SED potential. Specifically, we aim to (1) identify the locations most suitable for SED within the Southwest, (2) assess the habitat suitability of selected species, and (3) evaluate the approximate percentage of overlap between high SED potential and species habitat. This integrated analysis offers a practical pathway for balancing conservation goals with renewable energy expansion. To achieve this, we analyzed photovoltaic, environmental, and structural features to identify areas of high suitability for SED. We used species distribution modeling (SDM) to identify areas of high habitat suitability (hereafter hotspots of habitat suitability) within their known geographical range. SDMs, widely used in conservation planning (Addison et al., 2013), provided insights into high-priority habitats. Finally, we conducted an overlay analysis to assess the overlap between SED potential and priority habitats for the target species.

Methods and Materials

Study area

The study area encompasses the arid regions of the Southwest, spanning California, Nevada, Utah, Colorado, Arizona, New Mexico, and Texas (Fig. 1). It is rich in habitat and biodiversity, encompassing topographic extremes and wide climate diversity. Notably, it includes the country’s highest point, Mount Whitney, at 14,494 ft, and the lowest, Death Valley, at −282 ft (Southwest Climate Adaptation Science Center, 2023). Extreme topography variation can significantly influence various climatic parameters, such as temperature, precipitation, soil characteristics, and other ecological factors (Dillon et al., 2011). The Southwest is home to several ecosystems, including deserts, grasslands, woodlands, chaparral, tundra, wetlands, and various forested environments (Dahms & Geils, 1997). The region also supports extensive human activities and is home to some of the nation’s most productive agricultural land and urban areas, thus making this region an intriguing intersection of ecological complexity and human influence.

Species of greatest conservation need

The LeConte’s Thrasher (Toxostoma lecontei) and Bendire’s Thrasher (Toxostoma bendirei) are experiencing significant declines, making them among the fastest-declining species in the Southwest (Ammon et al., 2020). These birds are native to desert flats with sparse vegetation, such as saltbush, cholla cactus, and low shrubs (Sheppard, 2020; England & Laudenslayer Jr, 1993). Both species are particularly vulnerable to habitat alteration and loss due to their preference for specific vegetation types (Sheppard, 2020; England & Laudenslayer Jr, 1993). The risk is further exacerbated because SED projects often target landscapes with low-growing shrubs that can be easily removed during construction (Mierzwiak & Calka, 2017).

The Mojave Desert Tortoise (Gopherus agassizii) and the newly distinguished Sonoran Desert Tortoise (Gopherus morafkai) (Murphy et al., 2011) are biologically similar. Both species are characterized by their fossorial behavior; they construct burrows, creating microhabitats that provide shelter for themselves and many other desert inhabitants (Lovich & Ennen, 2011). The conservation status of G. agassizii is listed as critically endangered as it faces many threats, including those arising from renewable energy development (Berry et al., 2021).

The population of Burrowing Owl (Athene cunicularia) within the Southwest has steadily declined for many years, prompting conservation efforts to preserve this species (US Fish and Wildlife Service, 2023). The resident population within the Southwest is valuable to the long-term persistence of the species, considering its unique genetic diversity (Hughes, Daily & Ehrlich, 1998).

SED and SDM preparation

The combined approach to find areas of high habitat suitability for endangered species within areas of high SED potential included two major steps: (1) the use of abiotic variables to identify areas with the highest potential for SED and (2) the use of environmental variables and machine learning models to find suitable habitat of Species of Greatest Conservation Need.

Identifying sites with the highest potential for SED

We selected five variables often associated with solar energy development: global horizontal irradiance (Solargis, 2022), land cover (Doljak & Stanojević, 2017; Mierzwiak & Calka, 2017; Bukhary, Ahmad & Batista, 2018; Nebey, Taye & Workineh, 2020), slope (Charabi & Gastli, 2011; US Department of Energy, 2015; Alami Merrouni, Mezrhab & Mezrhab, 2016; Sabo et al., 2017; Rodrigues, Coelho & Pedro, 2017), proximity to substations (Katkar et al , 2021; Goh et al., 2022), and the exclusion of any listed protected areas (UNEP-WCMC & IUCN, 2023).

We obtained GHI from Solargis (2022). GHI represents the sum of direct and indirect diffuse solar irradiation received and is used as a first approximation of photovoltaic power production (ESMAP, 2020). We acquired the land cover data from the North American Land Change Monitoring System (Commission for Environmental Cooperation, 2023). To refine the analysis, we excluded land cover classes considered unsuitable for large-scale solar energy development: urban/built-up areas, wetlands, open water, forested areas, and snow/ice areas (Mierzwiak & Calka, 2017; Nebey, Taye & Workineh, 2020; Doljak & Stanojević, 2017; Bukhary, Ahmad & Batista, 2018). We finally reclassified each pixel representing suitable land classes as one. We obtained slope values from EarthEnv (Amatulli et al., 2018). We excluded slope values that exceeded five degrees through reclassification to focus on slope values suitable for solar energy development. The remaining slope values were assigned one value (Alami Merrouni, Mezrhab & Mezrhab, 2016; Charabi & Gastli, 2011; Sabo et al., 2017; Rodrigues, Coelho & Pedro, 2017). We acquired the substation data from the U.S Energy Atlas (Energy Information Administration, 2024). To estimate the distance between focal points, i.e., potential SED areas, we created a 5-mile buffer around each substation (Goh et al., 2022; Katkar et al , 2021). The resulting buffer zones served as a spatial indicator for identifying potential sites with high solar energy development potential. Next, we assigned each pixel within the buffer zone as one.

We obtained the protected areas from UNEP-WCMC and IUCN (UNEP-WCMC & IUCN, 2023). We excluded any areas classified as protected lands by the IUCN from our analysis. These lands included; strict nature reserves, wilderness areas, national parks, natural monuments or features, habitat/species management areas, protected landscape/seascapes, and protected areas with sustainable use of natural resources (Dudley, 2008; Stolton, Shadie and Dudley , 2013). This process allowed us to ensure that our study did not include any areas where SED would conflict with protected areas. By excluding these areas, we can focus our analysis on identifying sites with the highest potential for SED while minimizing negative impacts on the environment and sensitive species in the region. We assigned a value of one to the non-protected areas. All variables were upscaled to the same spatial resolution as the habitat suitability maps.

We overlaid the processed environmental and structural maps to calculate the areas with high SED suitability. Since the suitable areas for each fact were reclassified to one, we produced a summed map. We considered value four (100% overlap) to be suitable for environmental and structural areas –the SESA map. We overlaid the sites suitable for solar energy (SESA) map with the GHI map to mask SESA areas within the GHI map –sites suitable for potential SED development. Finally, we obtained the United States Large-Scale Solar Photovoltaic Database (USPVDB, Fujita et al., 2023), a national spatial database that provides the distribution of photovoltaic facilities with 1 megawatt or more. We overlaid the map of high SED suitability to the USPVDB to estimate the percent of USPVDB facilities included by our proposed solution.

SDM preparation and evaluation

Occurrence and environmental data.

We used occurrence data for the selected species obtained from GBIF (GBIF, 2023a; GBIF, 2023b; GBIF, 2023c; GBIF, 2023d), as well as solar and bioclimatic data from BioClim (WorldClim, 2020), and topography data from EarthEnv (Amatulli et al., 2018). To ensure high data quality and accuracy, the occurrence data underwent a thorough cleaning process, which included reducing spatial aggregation, removing duplicate occurrences and records with missing values, and incorporating pseudo-absences and background data. To mitigate the potential impacts of sampling bias and improve the quality control of the data, we first removed duplicate records and those with incomplete or inaccurate geographic positions (e.g., in the ocean). We then created a grid with the same resolution of the environmental variables to randomly select one per site (Hijmans et al., 2022).

Variable selection.

To minimize potential multicollinearity in the model, we used the VarSel function from the SDMTune package (Vignali et al., 2020) to remove highly correlated variables. This step helps to ensure that only variables with the most significant influence are included in the analysis, thereby improving the accuracy and reliability of the results. We generated 10,000 background locations using the dismo package in R (Hijmans et al., 2021). The data was then split into training and testing data sets, with a 20% allocation for testing. Then, a Maxtent model (Phillips, Dudik & Schapire, 2004) was employed, containing all variables (Elith et al., 2011). Maxent is successfully used to estimate species’ habitat suitability (Elith et al., 2011). The varSel function was applied to perform data-driven variable selection. Starting from the provided model, it iterates through all the variables, starting from the one with the highest contribution (permutation importance or maxent percent contribution). The method used for assessing variable correlation was Spearman’s rank correlation coefficient, and the threshold used was 0.7 (Vignali et al., 2020). The varSel function selects the least correlated variables based on the specified correlation threshold (Vignali et al., 2020). This process was performed for each target species.

Model evaluation & prediction.

First, we prepared presence and background locations and split the data into 80% for training and 20% for testing. Then, we used the subset of variables indicated by the variable selection process, the training dataset, and Maxent to estimate habitat suitability for each species. We restricted the produced habitat suitability to the known geographical ranges of the target species described by the IUCN (Figs. 2 and 3). Once we estimated habitat suitability, we applied the specificity/sensitivity threshold to transform suitability maps into binary (presence/background) (Liu et al., 2005). Lastly, we calculated the AUC (Area Under the Curve) and TSS (True Skill Statistics) values to evaluate each model. AUC and TSS are commonly used metrics for evaluating the performance of species distribution models, such as Maxent models (Elith et al., 2011). The AUC measures the model’s ability to distinguish between presences and absences. AUC scores range from 0 to 1, meaning higher AUC values indicate better model performance in determining suitable and unsuitable habitats (Elith et al., 2011). The TSS is another evaluation metric that combines specificity and sensitivity into a single statistic. TSS scores range from −1 to 1, meaning higher TSS values indicate better model performance in correctly identifying suitable and unsuitable habitats (Elith et al., 2011).

Figure 2 Habitat Suitability Mapping for Each Target Species within the IUCN resident extant.

Habitat suitability for T. lecontei, G. morafkai, and G. agassizii within their respective IUCN ranges.

Figure 3 Habitat suitability mapping for each target species within the IUCN breeding and resident extant.

Habitat suitability for both breeding and resident populations of T. bendirei and A. cunicularia within their respective IUCN ranges.

Identifying high-priority habitats

We calculated the overlap between the species presence produced by the SDM and the potential SED locations to identify overlapping areas. We considered areas where the potential SED locations and species presence had a value of 1 while assigning N.A. values to any cells with values other than 1. To express the overlap quantitatively, we calculated the percentage of overlap as the ratio of the sum of cells with overlapping values to the sum of cells with presence data (excluding N.A. values). The overlap percentage was reported as critical areas for SED development, i.e., areas where solar energy development could be pursued while considering the selected species’ habitat.

Results

SED analysis

The sites identified through the site suitability analysis are depicted in Fig. 4. These selected locations exhibit consistent traits, including a slope of less than five degrees, proximity within five miles of an existing substation, classification within one of four land cover categories (barren land, shrubland, grassland, or cropland), and a prominent GHI value ranking within the top 30%. The geographic distribution of these sites indicates that the sites suitable for potential SED development are mainly distributed in Texas (all over the state), California, and Colorado (Fig. 4). A major portion of suitable sites was also observed outside of urban areas of Arizona and Colorado. The overlay between the map of sites suitable for SED development and the USPVDB map was moderate, since 48% of USPVDB facilities were included in our proposed solution.

Figure 4 Spatial distribution of suitable sites for potential solar energy development (SED) in the Southwest Region.

This map identifies the geographic distribution of potential sites suitable for SED, focusing on key variables such as slope, land cover, proximity to existing substations, and Global Horizontal Irradiance (GHI). It serves as a predictive tool for guiding future renewable energy projects.

SDM results

The SDM outcomes for each target species are presented in Figs. S.1 and S.2. These figures show the habitat suitability and presence patterns for each species. These findings provide valuable insights into the specific variables that strongly influence the distribution and suitability of habitat for each target species. The high AUC and TSS values emphasize the reliability and accuracy of the SDM results. Values ranged from 0.89−0.98 for AUC and 0.63−0.91 for TSS (Athene cunicularia: AUC = 0.87, TSS = 0.6, Toxostoma lecontei: AUC = 0.98, TSS = 0.89: Toxostoma bendirei, AUC = 0.98, TSS = 0.89: Gopherus agassizii, AUC = 0.98, TSS = 0.89, Gopherus morafkai: AUC = 0.98, TSS = 0.92).

The variable response curves for A. cunicularia (Fig. S.6), highlight the species’ sensitivity to precipitation and solar radiation, which are key drivers of its distribution across California, Colorado, New Mexico, and Texas. For G. agassizii (Fig. S.7) and T. lecontei (Fig. S.9) temperature and solar radiation significantly influence their suitability, with high suitability patterns evident in California. Lastly, the response curves for G. morafkai (Fig. S.8) and T. bendirei (Fig. S.10) indicate a strong association with temperature and solar radiation, leading to a high suitability concentration in Arizona.

High-priority habitats

The geographic patterns of areas with projected presences for endangered species within regions of high SED potential are illustrated in Figs. 5 and 6. For the Burrowing Owl (A. cunicularia), the most critical areas are distributed in Texas, California, and Arizona. For LeConte’s Thrasher (T. lecontei), Bendire’s Thrasher (T. bendirei), the Mojave Desert Tortoise (G. agassizii), and the Sonoran Desert Tortoise (G. morafkai), the most critical areas are observed in California and Arizona (Figs. 5 and 6).

Figure 5 Overlap analysis: potential solar energy development (SED) sites and target species presence with IUCN breeding and resident extant.

Displays the estimated overlap percentages between the predicted species presence and SED overlap for T. bendirei and A. cunicularia within their respective breeding and resident IUCN ranges.

Figure 6 Overlap analysis: potential solar energy development (SED) sites and target species presence with IUCN resident extant.

Displays the estimated overlap percentages between the predicted species presence and SED overlap for T. lecontei, G. morafkai, and G. agassizii within their respective IUCN ranges.

Regarding the overlap between areas of projected presence and SED potential for non-migratory species, the highest values are observed for the Sonoran Desert Tortoise (G. morafkai, 46.36%) and LeConte’s Thrasher (T. lecontei, 35.81%), while the lowest is observed for the Mojave Desert Tortoise (G. agassizii, 16.69%).

For migratory species, the overlap of projected presences and SED potential for Bendire’s Thrasher (T. bendirei) is 31.44% for resident extant individuals who are present year-round and 6.38% for breeding individuals who migrate for a portion of the year. Similarly, for the Burrowing Owl (A. cunicularia), the overlap is 6.76% for resident extant individuals and 13.63% for breeding individuals. This distinction is important as it highlights the different conservation needs and potential impacts of SED projects on species with both migratory and resident populations.

Discussion

This study aims to connect the knowledge gap between the long-term implications SED may have on biodiversity by employing a multifaceted approach that combines established methodologies and decision-making processes. Specifically, we integrated the outcomes of a site suitability analysis and SDM, which are widely recognized and utilized within the scientific research community. Similar broad-scale research, such as the Arizona case study by Thomas et al. (2018), evaluated species richness across all vertebrate taxa, highlighting the overlap with Bureau of Land Management (BLM) variance areas and low conservation priority lands as identified by the Arizona Game and Fish (AZGF). Their research provides a valuable initial baseline of wildlife activity in these contested regions. Recently, (Karban et al., 2024) studied the potential impacts of SED on Wildlife and plants across deserts of the Southwest and they reported that species with broader ecological niches are less vulnerable to SED operations, while species with narrow niches are more likely to be affected by these SED activities. Our study complements these efforts by linking specific species data, including broad and narrow niche species, to SED It also offers insights into critical habitats and future renewable energy projects. This approach contributes to our understanding of renewable energy development and conservation planning.

SED suitability analysis

The site suitability analysis identifies potential regions for large-scale solar energy developments, including (a) California Central Valley: This region’s potential for future solar projects is primarily due to its relatively flat terrain and extensive agricultural land. The flat terrain facilitates easier installation and maintenance of solar panels. At the same time, agricultural land may offer large, open spaces suitable for SED (Doljak & Stanojević, 2017; Mierzwiak & Calka, 2017; Bukhary, Ahmad & Batista, 2018; Nebey, Taye & Workineh, 2020); (b) Greater Phoenix Valley, Arizona: The dense cluster of potential sites in this area is due to the combination of flat topography and high Global Horizontal Irradiance levels. High GHI levels indicate abundant solar radiation, which is necessary for maximizing the efficiency and energy output of solar panels (Gbémou et al., 2021; GlobalSolarAtlas, 2021; Doljak & Stanojević, 2017), (c) Texas: The analysis reveals suitable locations throughout Texas. The widespread potential in Texas is attributed to high GHI levels and ideal land cover, including shrublands, croplands, and grasslands. These land cover types are often well-suited for SED because they can provide large, relatively unobstructed areas for panel installation (Mierzwiak & Calka, 2017; Bukhary, Ahmad & Batista, 2018; Nebey, Taye & Workineh, 2020).

Our results, however, revealed a moderate overlap between areas of high suitability of SED and the US Large-Scale Solar Photovoltaic Database (USPVDB, Fujita et al., 2023). This moderate overlay could be linked to the lack of incorporation of datasets (i.e., residential, commercial and industrial PV facilities) representing the current distribution of operational solar facilities. We did not incorporate them, because these are not widely available at the scale of our study. Even with these limitations, our analysis offers a predictive framework, serving as a backdrop for conservation data. By identifying areas with SED potential the inclusion of biodiversity data can highlight conflict with sensitive habitats.

SDM results

Our findings indicate a significant correlation between habitat suitability for each of the five target species and climate and environmental variables, albeit with some variation. Notably, three key climatic and environmental variables exhibited the most substantial influence, consistently shared across all five target species. Precipitation emerged as a top influential factor for T. lecontei, T. bendirei, G. morafkai, and A. cunicularia. This correlation can be attributed to their dietary reliance on arthropods and small rodents, populations of which frequently correlate with precipitation levels and the overall health of vegetation (Desmond & Sutton, 2017; McDonald, Korfanta & Lantz, 2004). Temperature played a crucial role for T. lecontei, G. agassizii, and G. morafkai, with temperature shifts impacting activity levels, feeding patterns, and species survivorship (Meyer, 2008 Sheppard, 1970). Topographic variables like elevation and Topographic Roughness Index (TRI) significantly impacted T. bendirei and A. cunicularia, influencing habitat preferences, distribution, and adaptation to specific landscapes (Desmond & Sutton, 2017; Meyer, 2008).

The influence of these variables on the species’ habitat suitability emphasizes their critical importance in shaping the ecological conditions for each target species. Consequently, any significant or prolonged alterations in the environmental or climatic variables could directly affect the target species. For instance, precipitation’s impact on food source availability reveals concerns about these species’ ability to find sufficient sustenance, potentially hindering reproductive success and population health (Desmond & Sutton, 2017; McDonald, Korfanta & Lantz, 2004). Temperature’s influence on activity levels and survivorship highlights these species’ adaptations to cope with extreme heat conditions, with different responses seen in burrowing owls and desert tortoises (Meyer, 2008; Sheppard, 1970). The connection between topography and habitat preferences also showcases the importance of specific landscape characteristics for T. bendirei and A. cunicularia, influencing their presence in landscapes with features like exposed ground patches or open flat land (Desmond & Sutton, 2017; Meyer, 2008).

The Southwest has been identified as a climate change “hotspot,” with projected increases in air temperature, aridity, and seasonal variability (Gutzler & Robbins, 2011). Arid environments, such as deserts in the Southwest, are particularly vulnerable to the impacts of climate change, supported by abundant evidence (Archer & Predick, 2008; MacDonald, 2010; Garfin, 2013). For instance, the region is expected to experience fewer frost days, more frequent unusually high temperatures, increased water demand, and a higher frequency and intensity of extreme events such as droughts, heatwaves, and floods (Archer & Predick, 2008; MacDonald, 2010; Garfin, 2013). While changes in precipitation hold a higher degree of uncertainty, likely, both precipitation and temperature variations will directly impact vegetation and ecosystem processes throughout the Southwest (Archer & Predick, 2008). Nevertheless, it’s important to acknowledge that these findings are grounded in current data, and future changes in these variables due to factors such as climate change and future urban expansion may introduce additional complexities to these species’ habitat suitability and species welfare. Given the strong correlation between temperature, precipitation, and the overall fitness of target species, the suitable habitat is subject to change based on future environmental and climate changes.

Overlap results—implications for conservation

Our analysis considered habitat suitability and projected species presence to gain insights into the potential impacts of future SED locations. Species presence refers to documented occurrences of a particular species within a specific geographic area. At the same time, habitat suitability assesses the environmental conditions and resources necessary for a species to thrive and persist. It’s important to note that the percentages of species presence provided in our analysis were determined within the ranges specified by the IUCN, and they are approximations. Additionally, these percentages may change based on currently available data and the data used in our analysis is subject to change over time.

Target species and SED overlap estimations

The spatial overlap between potential SED sites and projected species presences underlines the significance of prioritizing species conservation and habitat preservation within the Southwest. Certain species, such as those studied here, overlap substantially with prospective SED locations, indicating potential risks to their habitats and populations. Considering the average percentages of overlap and variations among different species, it becomes evident that careful consideration and mitigation measures are necessary to balance renewable energy goals with preserving biodiversity and ecosystem integrity.

While the long-term impacts of large-scale SED remain a subject of ongoing research, a recent study by Bennun et al. (2021) provides crucial insights into some potential impacts. Their study highlights that habitat loss, resulting from vegetation clearance and ongoing facility maintenance and management, directly affects surrounding biodiversity. Additionally, their results highlighted additional adverse effects, including avian collisions with solar panels or transmission lines, the creation of barrier effects, and increased light and noise pollution (Bennun et al., 2021) Recognizing the interconnected challenges, it’s noteworthy that both habitat loss and fragmentation pose significant threats to terrestrial biodiversity (Rogan & Lacher, 2018).

Our results indicated that the highest overlap between projected presences and SED was observed for the G. morafkai, a vulnerable species whose population is decreasing because of habitat degradation and severe habitat fragmentation (Averill-Murray et al., 2023). The Sonoran Desert Tortoise relies on desert habitats for food, water, protection, and breeding (Averill-Murray et al., 2023). As the SED expands into suitable areas, it may limit the tortoise dispersal. When habitat is impacted by anthropogenic activities (e.g., urban infrastructure) or natural disasters, the landscape may become broken into pieces of unconnected habitats (Berger-Tal & Saltz, 2019). If no measures are taken to conserve the remaining suitable habitat of the G. morafkai, individuals may have difficulty accessing food or water resources. The outputs from this research can help managers and conservationists target suitable habitats, secure food and water resources, and help individuals move among fragmented habitats. The proposed solution can also be applied to least concerned species, such as T. lecontei. Human modifications to the landscape are considered a major driver of the decline in the distribution and abundance of non-threatened species (Baker et al., 2019). The maps produced herein can help to prioritize suitable sites to prevent habitat fragmentation and reconnect isolated patches.

By addressing areas of overlap, we inherently identify regions where SED development can potentially occur without significant conflict. However, when refraining from SED in overlapping areas is unfeasible, mitigation becomes essential. Figure S.3 exemplifies the availability of potential SED sites while still considering the priority habitat of the target species. Mitigation strategies can take various forms to overcome potential habitat loss and fragmentation problems, including proactive measures during SED project design and operational phases. Project design adjustments, such as altering SED layouts to avoid critical migratory corridors and nesting/roosting sites for specific species or rerouting and burying power lines to reduce avian collisions, play a crucial role in minimizing environmental impacts (Bennun et al., 2021). Additionally, operational mitigation efforts encompass modifying perimeter fencing to minimize barrier effects by creating passageways for smaller species (Bennun et al., 2021).

Our results can also help to implement new policies to mitigate potential impacts of habitat fragmentation on biodiversity. A recent study by Karban et al. (2024) emphasized the application of the National Environmental Policy Act (NEPA) mitigation hierarchy, which outlines a tiered approach for minimizing the impacts of SED on sensitive resources, including species. The hierarchy consists of three levels: avoidance, minimization, and restoration or offsetting. The maps provided herein can be used to assess policy priorities regarding the inclusion of the aforementioned levels in conservation planning. While promising, these mitigation strategies highlight the need for further species-specific research in the application of SED, given the diverse range of taxa in the Southwest.

Limitations

While our study provides valuable insights into the overlap between high-priority habitats and suitable locations for SED, certain limitations must be acknowledged. First, the site suitability analysis relies on current substation data without accounting for future substation or transmission line availability. Additionally, ground truthing with existing solar data is needed to evaluate the areas of SED better, as commercial and residential solar data were excluded from the analysis due to a lack of open access. Including this data could offer important insights for future studies, and its absence may affect the long-term viability and feasibility of the identified SED sites. We acknowledge that the SED model can be further refined to enhance its predictive accuracy and better capture the complexities of SED suitability.

Secondly, the focus on the species’ range within the United States Southwest, while providing valuable information, may not fully capture the complete spatial distribution or habitat suitability, as the range of each target species extends beyond our study area.

Furthermore, it’s important to recognize that our spatial overlap analysis provides an approximation of the interaction between the potential SED sites and the target species’ habitat. The actual spatial dynamics at a finer scale may differ, and the species’ habitat suitability could change over time due to factors like climate change, impacting distribution and habitat availability. It is also important to note that the percentages of species presence and habitat suitability provided in our study were determined within the ranges specified by the IUCN, and they are approximations. Additionally, these percentages may be subject to change based on currently available data, and the data used in our analysis will be subject to change over time.

Although site suitability analyses are standard in research and planning, our inclusion of species-specific data provides added ecological insight by highlighting interactions between future SEDs and high-priority habitats. We envision that similar analyses expanding to a broader range of taxa could prove valuable in guiding future conservation decisions. By recognizing these limitations, we aim to inspire further studies and conservation efforts that actively address potential changes and uncertainties tied to substation availability and environmental conditions.

Conclusion

As countries transition to renewable energy sources to combat the escalating climate crisis, a growing focus is on identifying suitable areas for renewable energy development, such as SED. Multiple studies have identified the Southwest as undoubtedly the most ideal location for SED in the United States. However, the potential long-term implications on biodiversity remain understudied in this pursuit of renewable energy.

The importance of mitigating the impacts of SED on surrounding ecosystems and wildlife cannot be overstated. Our primary objectives of this study included identifying highly suitable locations for potential SEDs in the Southwest, identifying suitable habitats for the target species, understanding the intricate relationship between habitat suitability and environmental variables, and identifying suitable SED sites within each species’ range. By employing diverse methodologies, such as site suitability analysis and species distribution modeling, to identify shared critical areas, we hope this information can help inform future conservation and decision-making in the transition towards sustainable and renewable energy.

Supplemental Information

Supplemental Information 1 The code used in R, and the occurrence data for each selected species

Supplemental Information 2 Habitat suitability mapping for each target species

The habitat suitability of five target species is illustrated —Gopherus morafkai , Gopherus agassizii , Toxostoma bendirei , Athene cunicularia , and Toxostoma lecontei —across the Southwest. Habitat suitability values range from 0 (low suitability) to 1 (high suitability).

Supplemental Information 3 Predicted presence for each target species

The predicted presence of the five selected species in the Southwest based on species distribution modeling (SDM). Presence data are aggregated to depict regions with a high likelihood of species occurrence.

Supplemental Information 4 SED opportunities when considering biological impact

Orange areas denote regions identified as high priority habitats for the selected species. Yellow regions represent suitable locations for potential SED.

Supplemental Information 5 IUCN ranges

Supplemental Information 6 Supplementary information detailing steps taken throughout the analysis

This flowchart outlines the workflow used in the study to combine site suitability analysis with species distribution modeling. The diagram details each analytical step, including variable selection, habitat suitability modeling, and overlap analysis, providing a comprehensive view of the methodology and decision making.

Supplemental Information 7 A. cunicularia variable model response

The variable response curves of key environmental variables influencing the habitat suitability for Athene cunicularia. Each curve represents the relationship between environmental such as precipitation, solar radiation.

Supplemental Information 8 G. agassizii variable model response

The variable response of Gopherus agassizii to climatic and environmental predictors. The model identifies influential variables such as temperature and solar radiation, showing how variations in these factors can affect the predicted habitat suitability.

Supplemental Information 9 G. morafkai variable model response

The variable response model for Gopherus morafkai, showing the influence of key variables, such as temperature and solar radiation, on habitat suitability. The response curves reflect the sensitivity of the species to changes in environmental conditions.

Supplemental Information 10 T. lecontei variable model response

The variable response model for Toxostoma bendirei , demonstrating how environmental factors, including temperature and solar radiation, influence habitat suitability.

Supplemental Information 11 T. bendirei variable model response

The variable response model for Toxostoma bendirei, demonstrating how environmental factors, including temperature and solar radiation, influence habitat suitability.

We thank the editor, Liz Kalies, and two anonymous reviewers for their comments and suggestions.

Additional Information and Declarations

Competing Interests

Author Contributions

Data Availability

The authors declare there are no competing interests.

Kylee Fleckenstein conceived and designed the experiments, performed the experiments, analyzed the data, prepared figures and/or tables, authored or reviewed drafts of the article, and approved the final draft.

Adam Stein conceived and designed the experiments, analyzed the data, authored or reviewed drafts of the article, and approved the final draft.

Heather L. Bateman conceived and designed the experiments, analyzed the data, authored or reviewed drafts of the article, and approved the final draft.

Fabio De Albuquerque conceived and designed the experiments, performed the experiments, analyzed the data, prepared figures and/or tables, authored or reviewed drafts of the article, and approved the final draft.

The following information was supplied regarding data availability:

The code, GIS variables and dataset are available in the Supplemental Files.

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
