# Peer review of "Mapping species of greatest conservation need and solar energy potential in the arid Southwest for future sustainable development"

_PeerJ, doi:10.7717/peerj.18568_

## Round 0.1 · original submission · Minor Revisions

Dear Authors,

Thank you for the excellent work in addressing the reviewers' comments. However, some minor revisions are still required, particularly Experimental design and Validity of the findings, to bring the manuscript to an acceptable standard for publication.

We greatly appreciate your choice of PeerJ for submitting such an engaging and valuable manuscript.

Best regards,
Armando Sunny

Reviewer 1 ·

Basic reporting

Needs clarity as indicated in comments below

Experimental design

Seems sound

Validity of the findings

Conclusions are reasonable

Additional comments

This is an interesting paper although the approach used has been implemented before (see Thomas et al. 2018 below). The manuscript is generally well-written and the topic is relevant and timely. It appears that the authors are not completely familiar with the literature on the topic of solar energy development and wildlife so I provided citations they should examine and consider citing. There are probably other citations as this is a rapidly growing field of inquiry.

I have two main concerns with the paper. The first is the use of endemic to describe the species they include. Endemic to where? The huge region they selected, a state (they imply Arizona), or something else. I’m not sure why they selected the species they did and not others that I listed in a comment on the ms that would appear to be good candidates. This matter needs to be addressed in the revision.

Second is their frequent reference to the model applying best to species in the southwest. That seems like circular reasoning given that that is the area they selected in their analysis. Maybe I misunderstood something but if so it needs to be clarified.

I made other comments on the ms.


AGHA, M., J. E. LOVICH, J. R. ENNEN, AND B. D. TODD. 2020. Wind, sun, and wildlife: do wind and solar energy development “short-circuit” conservation in the western United States? Environmental Research Letters. 15.
KARBAN, C. C., J. E. LOVICH, S. M. GRODSKY, AND S. M. MUNSON. 2024. Predicting the effects of solar energy development on plants and wildlife in the Desert Southwest, United States. Renewable and Sustainable Energy Reviews. 205:114823.
MOORE-O'LEARY, K. A., R. R. HERNANDEZ, D. S. JOHNSTON, S. R. ABELLA, K. E. TANNER, A. C. SWANSON, J. KREITLER, AND J. E. LOVICH. 2017. Sustainability of utility-scale solar energy – critical ecological concepts. Frontiers in Ecology and the Environment. 15:385-394.
THOMAS, K. A., C. J. JARCHOW, T. R. ARUNDEL, P. JAMWAL, A. BORENS, AND C. A. DROST. 2018. Landscape-scale wildlife species richness metrics to inform wind and solar energy facility siting: An Arizona case study. Energy Policy. 116:145-152.

Annotated reviews are not available for download in order to protect the identity of reviewers who chose to remain anonymous.

·

Basic reporting

The introduction needs more review of the current body of literature around solar development and wildlife interactions, which is not a huge body of literature, but relevance can be pulled from other regions of the country. In addition, a thorough review of studies from the American Southwest is needed.
Similarly, there are a lot of published studies on solar suitability, and a paragraph that summarizes these for the US, particularly for the Southwest, is needed. Please explain what data gaps exist that make your study a contribution. For example, most existing solar suitability studies are pretty light on conservation data, so that is an easy point to make.
Figures 3-6- I vote to use common names instead of scientific.
Figure 2 should be introduced before Figs 3 and 4.
As noted in the PDF, "the Southwest" is capitalized but "southwestern region" is not.
The highlighted sentence on page 14 refers to Figs 5 and 6 as habitat suitability, but I think those figures are showing presence? Or are areas of presence also considered "high" habitat suitability? Be careful of this distinction throughout the results/discussion, it gets a bit confusing.
Would it be possible to put the results listed in 4.2 on environmental and climatic variables in a table (species vs variables)? Fine to put it in the supplementary material, but these results would be good to see (and should be presented in results, not discussion).

Experimental design

Good detail on methods.
I don't know for sure if the IUCN database is inferior to the USGS PAD-US, but I would check out the latter because I think a US, government-maintained database on legally protected lands might be better than a global one (although possibly one feeds the other).
On a related note, please explain how you dealt with different levels of "protection." Some lands, like national parks, are totally off limits to development. BLM, however, has a permitting process that allows development in designated areas after a lengthy review. I'm curious how you handled this, because it gets complicated in this region to designate areas as protected or unprotected in a binary way.
You modeled and buffered substations as a way of identifying areas close to transmission, but the analysis might be strengthened if you included the grid (transmission lines) where solar could be located, effectively increasing the SED. You might want to look at (and cite) other solar suitability models to compare methods.

Validity of the findings

Your SED analysis (places likely suitable for solar development) would be easy enough to groundtruth against existing solar build-out; please consider adding this to the analysis. The results seem reasonable, but I would be much more convinced if you showed that >90% of existing solar is within your footprint, for example.
The context and discussion on resident/migratory and breeding habitats was very nice.
The first paragraph of the discussion is a rehash of what you did- not needed.
In 4.3.1, is there any way to interpret the percentages themselves and talk about whether they are acceptable or too much overlap? I know that is a tough question, and maybe the better outcome is to compare and contrast between species to see who is most at risk. This could use more discussion, and relates to some of my concerns about applicability of the study to conservation efforts.
I like the discussion of avoid vs mitigation. It seems like you could beef up this section with more existing literature on solar-wildlife impacts, even if from other regions. Bennun 2021 is cited quite a bit and there is other literature on these topics.

Additional comments

In 4.1, I don't think the identification of potential solar sites is a novel or sophisticated contribution. I see the SED as a coarse analysis that is appropriate for your purposes, which is the overlay with listed species. There are more sophisticated analyses available on solar suitability, e.g., the Princeton Net Zero study. Please cite some of this literature and show your methods are comparable. Or, if I'm missing something and your analysis does represent an improvement over existing solar suitability models, please cite/explain. This relates to needing more in the intro on existing solar/environmental suitability lit review.

I was looking forward to the conclusion talking a bit more about how to make these results actionable, but on-the-ground or policy implications were not discussed. I think it would be good to suggest some next steps, or further explain the significance of your results for the renewable energy transition and conservation. It is hard to know how to interpret your results from the perspective of a conservation practitioner.

Reviewer 3 ·

Basic reporting

The paper is clear and unambiguous and with sufficient context. Also, the structure is correct, figures are mostly correct. Raw data is shared, although it should the code needs improvements by including it into a R project. Scientific name of species are sometimes incorrect spelled ...Toxostoma Lecontei should be Toxostoma lecontei, including in figures.

Experimental design

The methodology is not entirely clear. For example, the authors use IUCN ranges to restrict the analysis. This might be very restrictive, and I really don't understand why is this restriction. IUCN range is biased, as I would prefer a model for the entire study area, or using MCP as range substitution. Also, dividing the range into resident and breeding is not necessary and clutter the results. Methods for SDMs should be more clear (e.g., a table with variables for each species and justification for inclusion). For results there is a need for a table with variable influence score. Being a methodology paper, flowchart have to be in the paper not SI.

Validity of the findings

Discussion should be more on the methodology, and especially on how to integrate it into planning. Also, how can be this method scaled to other species, and what is his value for species with limited data.

Additional comments

no comment

---

## Round 0.2 · Minor Revisions

Dear Authors,

After reviewing the corrections made to your manuscript, I have found that some minor revisions are still necessary.

While you did address the request for an expanded literature review, only one additional citation was added. My suggestion for a more comprehensive review, particularly in the area of solar suitability, has not been adequately addressed. I encourage you to expand this section further to include more relevant references and a broader discussion on the topic.

Additionally, some of my requests for minor additional analyses were not fulfilled, with the explanation that these could not be completed. While I understand certain limitations, a more concerted effort in attempting to address these points would have been appreciated.

Please revise the manuscript accordingly and provide the necessary updates.

Sincerely,

Armando Sunny

·

Basic reporting

I don't think my comments on the intro and lit review were adequately addressed.
I recommended a literature review about solar impacts to species in the Southwest, which would help the reader understand what is currently known about wildlife impacts.

Similarly, I don't see a discussion on other solar suitability studies, whether conservation values have been considered in other solar suitability studies, or how this advances the existing literature.
I see that the factors you included for predicting solar are cited, but that isn't what I'm asking. I want to see how this research is different from other solar suitability studies.

The correction to "the Southwest" vs "southwestern US" is a minor point but I still see this error.

Experimental design

No comments, thank you

Validity of the findings

I mentioned that it would be great to see the current solar build-out compared to your modeled SED. I was surprised that you can't do this due to data limitations. Are you familiar with the USGS solar database? I'd like to reiterate that this would be good to include (I am not suggesting that you include distributed solar).
https://eerscmap.usgs.gov/uspvdb/

My other comment was addressed nicely.

Additional comments

I still think that the SED is a coarse analysis, which is very appropriate as a backdrop for your species analysis.
I maintain that you are overstating the value of the SED in section 4.1, by implying that it provides novel information for siting solar. This simply isn't necessary- there are better analyses out there for siting solar, and yours is perfectly appropriate for discussing potential biodiversity conflicts. I would delete most of 4.1 and simply talk about the accuracy of this analysis, hopefully compared to real solar facilities or other analyses.

I appreciate the inclusion of conservation implications!

Reviewer 3 ·

Basic reporting

Clear

Experimental design

Not very clear to me. The answers to1st review were all in all cases ... this is great suggestion for a future study. However the methodology is rigorous and with high ethical study.

Validity of the findings

Clear

Additional comments

The paper can be accepted as it is.

---

## Round 0.3 · accepted · Accept

Dear Authors

We are pleased to inform you that your manuscript, titled "Mapping species of greatest conservation need and solar energy potential in the arid Southwest for future sustainable development", has been accepted for publication in PeerJ.

Thank you for choosing PeerJ as the platform to share your research. We look forward to seeing the positive impact of your findings in the academic community.

Best regards,
Armando Sunny

·

Basic reporting

I am satisfied with the revisions

Experimental design

I am satisfied with the revisions

Validity of the findings

I am satisfied with the revisions

Additional comments

I am satisfied with the revisions